# Genetic Variants of HOTAIR Associated with Colorectal Cancer: A Case-Control Study in the Saudi Population

**DOI:** 10.3390/genes14030592

**Published:** 2023-02-26

**Authors:** Haya Saad Alzeer, Jilani P. Shaik, Narasimha Reddy Parine, Mohammad Alanazi, Abdullah Al Alamri, Ramesa Shafi Bhat, Sooad Al Daihan

**Affiliations:** 1Department of Biochemistry, College of Science, King Saud University, P.O. Box 22452, Riyadh 11495, Saudi Arabia; 2Genome Research Chair, Department of Biochemistry, College of Science, King Saud University, P.O. Box 22452, Riyadh 11495, Saudi Arabia

**Keywords:** HOTAIR, CRC, miR-141, SNP, lncRNA

## Abstract

Genetic polymorphism in long noncoding RNA (lncRNA) HOTAIR is linked with the risk and susceptibility of various cancers in humans. The mechanism involved in the development of CRC is not fully understood but single nucleotide polymorphisms (SNPs) can be used to predict its risk and prognosis. In the present case-control study, we investigated the relationship between HOTAIR (rs12826786, rs920778, and rs1899663) polymorphisms and CRC risk in the Saudi population by genotyping using a TaqMan genotyping assay in 144 CRC cases and 144 age- and sex-matched controls. We found a significant (*p* < 0.05) association between SNP rs920778 G > A and CRC risk, and a protective role of SNPs rs12826786 (C > T) and rs1899663 (C > A) was noticed. The homozygous mutant “AA” genotype at rs920778 (G > A) showed a significant correlation with the female sex and colon tumor site. The homozygous TT in SNP rs12816786 (C > T) showed a significant protective association in the male and homozygous AA of SNP rs1899663 (C > A) with colon tumor site. These results indicate that HOTAIR can be a powerful biomarker for predicting the risk of colorectal cancer in the Saudi population. The association between HOTAIR gene polymorphisms and the risk of CRC in the Saudi population was reported for the first time here.

## 1. Introduction

Cancer, in general, is the second leading cause of death in the world, and colorectal cancer is in third place after lung and breast cancer. In Saudi Arabia, the occurrence of colorectal cancer is on top after breast cancer in females but is the topmost common cancer in males [1]. The mechanism involved in the development of CRC is very complicated and both genetic and environmental aspects are considered risk factors [2]. Although the exact cause behind it is still not known, its occurrence is associated with nonmodifiable risks such as age, gender, genetics, and some modifiable factors including environment and lifestyle [3,4,5]. In the human genome, single nucleotide polymorphisms (SNPs) are considered the major genetic variants which are used to predict cancer risk and prognosis [6,7]. Almost 10 million SNPs are reported with a frequency of 1 in 300 nucleotides in the genome. SNPs located in the genes that regulate metabolism, immunity, or cell cycle regulation are usually associated with genetic susceptibility to cancer [8]. Understanding the relationship between SNPs and cancer susceptibility points toward the molecular pathogenesis of various cancers. SNPs are probably considered prospective diagnostic and therapeutic biomarkers in cancer [8]. In a gene, SNPs can be located at promoter regions, exons, introns, or even at 5′- and 3′ UTR, so they can alter the gene expression [9,10,11,12,13]. SNPs present in long noncoding RNAs (lncRNAs) are reported to be associated with cancer risk as they may change the structure and expression levels of lncRNA [14].

Generally, lncRNAs are RNA molecules of 200 nucleotides or more and mostly regulate protein expression and do not encode any protein. These molecules exist throughout the genome and interact with DNA, RNA, and protein to regulate protein expression through epigenetic, transcriptional, and post-transcriptional regulations [14,15,16]. As a key regulator of gene expression, the impairment of lncRNAs results in the prognosis, metastasis, and recurrence of different types of cancer. Recently, HOX transcript antisense intergenic RNA (HOTAIR) lncRNA, which is one of the most important regulatory RNAs in humans, has shown an association with cancer metastasis, chemotherapy responses, and the survival rate of patients [17]. Additionally, genetic polymorphism in HOTAIR is linked with the risk and susceptibility of human cancer [18]. Many studies have reported the abnormal expression of HOTAIR in cancer tissue. The overexpression of HOTAIR activates the gene-silencing pathways activated by modified histone protein. In cancer tissue, HOTAIR acts as a molecular decoy for microRNAs (miRNAs) and RNA-binding proteins (RBPs) and directly regulated the target mRNA [19]. HOTAIR suppresses miR-148a by competing with endogenous RNA in esophageal and epithelial cancer to promote the expression of Snail2 and to enhance cell invasion and metastasis via the epithelial-to-mesenchymal transition [20,21]. miRNA-34a is downregulated by HOTAIR in colon cancer and HOTAIR itself is upregulated in esophageal squamous cell carcinoma [22,23,24,25,26].

Studies have found worse prognosis in patients with upregulated HOTAIR in primary tumors or blood as compared to the patients with low HOTAIR expression, and thus scholars have proposed it as a potential biomarker [19,27]. In CRC cells, HOTAIR is found to reduce the expression levels of E-cadherin, which results in increased levels of vimentin and matrix metalloproteinase 9 (MMP-9) involved in the invasion and metastasis [28]. Several studies have reported the association between HOTAIR genetic polymorphisms and CRC risk [19,29,30]. Several SNPs of HOTAIR act as potential cancer susceptibility loci but no studies on the association of SNPs in the Saudi population has been reported. The present study reported three HOTAIR polymorphisms (rs920778 G > A, rs12816786 C > T, and rs1899663 C > A) to evaluate the association between HOTAIR variants and CRC prevalence in the Saudi population. These SNPs were selected on the basis of some recent reports linking them with increased cancer risk. Few molecular epidemiological studies linked HOTAIR rs920778 polymorphism with the risk of breast, cervical, and lung cancer; rs1899663 with lung, breast, and gastric cancer; and rs12816786 with lymphoma [31,32,33,34,35].

## 2. Materials and Methods

### 2.1. Subject Requirement

In the present case-control study of 288 individuals, 144 individuals with confirmed CRC cases were enrolled with an equal number of individual controls matched with age and sex from King Khalid University Hospital (KKUH) in Riyadh, Saudi Arabia. The study was approved by the ethical committee of KKUH. Informed consent from all the subjects was obtained.

### 2.2. Sample Collection

A total of 4 mL of peripheral blood was collected in ethylene diamine tetra acetic acid (EDTA) tubes (Hebei Xinle Sci & Tech Co. Shijiazhuang, China) from all the participants, and genomic DNA was isolated by using a Mini Kit from QI Aamp^®^. The purity and concentration of the extracted nucleic acids were quantified from the OD ratio (A260/280 nm).

### 2.3. SNP Selection and Genotyping

Three SNPs (rs920778, rs1899663, and rs12826786) in the oncogene lncRNA HOTAIR on Chr. 12 were examined. Genotyping was performed using TaqMan^®^ Genotyping and was analyzed using Quantstudio-7 (Applied Biosystems^®^, Life Technologies™, Carlsbad, CA, USA). The TaqMan^®^ genotyping reaction mix (10.2 µL/reaction) contained 8.2 µL of TaqMan genotyping master mix and 2 µL of a 20 ng/µL genomic DNA sample in 96-well plates according to the manufacturer’s instructions. The reactions were performed using the optimal thermocycler conditions for each target SNP: 30 s at 60 °C (preread Stage), 10 min at 95 °C (hold stage), then PCR stages 40 cycles of 15 s at 95 °C (denaturation) and 1 min at 60 °C (annealing), then the postreading stage of 30 s at 60 °C. The experiments were performed in triplicate as a quality control measure to verify the genotyping procedure.

### 2.4. Statistical Analyses

The data underwent statistical analysis by using IBM SPSS (Statistics for Windows, Version 23.0. IBM Corp, Armonk, NY, USA) and Microsoft Excel^®^. All reported *p*-values were two tailed and a *p*-value less than 0.05 was specified as statistically significant. The differences in the demographic variables and genotypes of the HOTAIR polymorphic variants (SNPs) (rs920778, rs1899663, and rs12826786) between the CRC cases and healthy controls were evaluated using the chi-squared (χ2) test. The Hardy–Weinberg equilibrium (HWE) was tested with a (χ2) test to study the likelihood of inheriting these HOTAIR (SNPs) into the Saudi population by comparing the results of the frequencies of the expected genotypes with the frequencies of the observed genotypes. The odds ratio (OR) was used to estimate the degree of correlation between HOTAIR genetic variation (SNPs) and the risk of CRC development. The 95% confidence interval (95% CI) indicates that the average of the actual value should be within the range. The 95% CI was considered significant if the difference range between the two values was less than 1. The OR and 95% CI were calculated using an (IHG) web tool (https://ihg.gsf.de/cgi-bin/hw/hwa1.pl, accessed on 6 February 2023). The RNAsnp Web server (https://rth.dk/resources/rnasnp/, accessed on 6 February 2023was used to predict the secondary structure of the HOTAIR variants with each SNP. A linkage disequilibrium (LD) analysis determines the degree of nonrandom associations between HOTAIR (SNPs). The LD analysis was carried out using Haploview.

## 3. Results

The clinical data of all the samples are summarized in Table 1. The CRC patients had a median age of 57 years, and the samples were divided into two groups: above 57 years old (*n* = 76, 52.7%) and below 57 years old (*n* = 68, 47.3%), with 86 males (59.7%) and 58 females (40.3%). In total, 91 patients (63.2%) were diagnosed with colon cancer, while 53 patients (36.8%) had rectal cancer. The stages of CRC included in this study included early stages (I-II) and late stages (III-IV); 61 (54%) of CRC patients had early stages of the disease, while 51 (46%) had a late-stage disease.

Table 2 illustrates the relationships between the genotype and CRC susceptibility, allele frequencies, and the significance of the genotype and allele distribution of several SNPs. Among the three HOTAIR SNPs studied, rs12826786 (C > T) and rs1899663 (C > A) demonstrated a statistically significant protective association (decreased odds ratio) in the Saudi CRC patients, while a third genotype rs920778 (G > A) demonstrated a statistically significant risk association (increased odds ratio) in the Saudi CRC patients. The mutant-homozygous genotype “AA” of rs920778 showed a significant risk correlation (OR (95% CI): 2.057 (1.063–3.981); χ2 = 4.64; *p* = 0.03131), and a minor allele “A” demonstrated a significance risk correlation (OR (95% CI): 1.438 (1.035–1.998); χ2 = 4.71; *p*= 0.02994). The additive genotype “GA + AA” showed a risk association with CRC but was not significant (OR (95% CI): 1.557 (0.931–2.606); χ2 = 2.86; *p* = 0.09078). The mutant-homozygous genotype “TT” for rs12826786 showed a significant protective correlation (OR (95% CI): 0.276 (0.074–1.020); χ2 = 4.18; *p* = 0.040) in the CRC patients, as did the additive genotype “CA + AA” of SNP rs1899663 (C > A) (OR (95% CI): 0.305 (0.131–0.710); χ2 = 8.21; *p* = 0.00417).

In the present study, the median age of the CRC and control subjects was 57. To estimate the association of HOTAIR SNPs with both CRC and the control patients, samples were divided into two groups (patients ≤ 57 years old and patients > 57 years old). The genotype frequencies of both groups are shown in Table 3. The SNP rs1899663 (C > A) showed a significant protective association in the CRC patients younger than 57 years old. The heterozygous variant “CA” at rs1899663 showed a significant protective association in CRC patients younger than 57 years old (OR: 0.231; χ2 = 5.9; CI: 0.066–0.801; *p* = 0.0151), as did the additive genotype “CA + AA” (OR: 0.28; χ2 = 4.8; CI: 0.085–0.926; *p* = 0.0289). The HOTAIR SNP rs920778 (G > A) was found to have a major risk association in CRC patients aged 57 years old. In CRC patients aged 57 years old, the homozygous “AA” mutant frequency was 3.3-fold higher than in healthy people (OR: 3.3; χ2 = 4.13; CI: 1.015–10.733; *p* = 0.043).

Table 4 showed the correlation of HOTAIR SNPs with gender. The mutant-homozygous genotype “TT” at rs12826786 was shown to have a protective association in male CRC patients (OR: 0.23; χ2 = 3.81; CI: 0.047–1.127; *p* = 0.051). On the other hand, a statistically significant risk association was observed in females with rs920778 (G > A) as shown in Table 4. The frequency of the homozygous mutant “AA” was increased 3.282-fold in female CRC patients compared to healthy subjects (OR: 3.282; χ2 = 4.91; CI: 1.129–9.536; *p* = 0.02669). The frequency of the rs920778 minor allele “A” was 1.768-fold higher in female CRC patients compared to controls (OR: 1.768; χ2 = 4.8; CI: 1.060–2.949; *p* = 0.0284).

Furthermore, the correlation of HOTAIR SNPs with tumor location was studied. Samples were classified into two groups according to the location of the tumor, either in the colon or the rectum. Remarkably, rs920778 showed a significant association with tumors located in the colon, while there was no correlation between any SNPs and tumors localized in the rectum (Table 5). The homozygous genotype “AA” at rs920778 in the patients with colon cancer showed 2.3-fold more significant risk compared to healthy individuals (OR: 2.332; χ2 = 4.75; CI: 1.082–5.027; *p* = 0.02926). The additive alleles “GA + AA” genotype showed significant association in the CRC patients compared to healthy individuals (OR: 1.895; χ2 = 4.24; CI: 1.027–3.497; *p* = 0.03940). The frequency of the minor allele “A” also showed significantly more correlation in colon cancer patients when compared to the control group (OR: 1.519; χ2 = 4.84; CI:1.046–2.206; *p* = 0.02785) (Table 5). Based on the results shown in Table 5, no association was observed between the HOTAIR SNPs examined (rs920778, rs1899663, and rs12826786) and rectal cancer patients.

The correlation between HOTAIR SNPs and the staging of colorectal tumors was studied. The stages of CRC were grouped into early stages (I-II) and late stages (III-IV) as shown in Table 6. The CRC patients with early-stage tumors showed a significantly higher risk (2.6 fold) with the homozygous variant genotype “AA” at rs920778 when compared to healthy people (OR: 2.615; χ2 = 4.73; CI: 1.085–6.304; *p* = 0.029). The genotype of additive “GA + AA” alleles showed a significant association in CRC patients compared to healthy individuals (OR: 2.042; χ2 = 3.86; CI: 0.994–4.195; *p* = 0.0493). The frequency of the minor “A” allele also showed a significantly higher association in the CRC patients compared to healthy controls (OR: 1.604; χ2 = 4.76; CI: 1.047–2.455; *p* = 0.0292) while the SNPs (rs1899663 and rs12826786) did not show any significant correlation with the CRC patients. The homozygous variant genotype “AA” at rs1899663 showed a significant protective association in late-stage CRC tumors compared to healthy individuals (OR: 0.344; χ2 = 4.11; CI: 0.119–0.993; *p* = 0.0426). However, none of the other HOTAIR SNPs studied (rs920778 and rs1899663) showed a significant association in late-stage CRC tumors.

A linkage disequilibrium (LD) analysis was performed to identify the LD between the SNPs. The LD blocks showed a very low association among the analyzed SNPs (Figure 1). The red color indicates a higher D’ value. The selected SNPs showed higher D’ values in the controls. All three HOTAIR SNPs studied showed an r2 = 1 value in association with other SNPs. The r2 values indicate that these loci are coinherited in LD.

Secondary structures of HOTAIR and base-pair probabilities were detected using an RNA Web server as shown in Figure 2. The RNAsnp predicted that a mutation at rs920778 would change the RNA secondary structure of the HOTAIR lncRNA. The RNAsnp predicted that the rs920778 G > A allele substitution would result in an MFE of −111.10 to −110.80 kcal/mol. The base-pair probabilities of the rs920778 wild type G allele and variant A allele were also different. The RNAsnp predicted that the rs12826786 C > A allele substitution would result in an MFE of −157.70 to −156.20 kcal/mol. The base-pair probabilities of the rs12826786 wild type C allele and variant A allele were also different. The RNAsnp predicted that the rs1899663 C> T allele substitution would result in an MFE of −120.90 to −121.80 kcal/mol. The base-pair probabilities of the rs1899663 wild type C allele and variant T allele were not significantly different (*p*-value = 0.0978).

## 4. Discussion

HOTAIR has gained widespread recognition as a functional lncRNA involved in a number of malignancies. It is situated on chromosome 12 within the Homeobox C gene cluster [36]. Research has shown that HOTAIR interacts with epigenetic regulators such as the lysine-specific demethylase 1A (LSD1) and polycomb repressive complex 2 (PRC2) complexes to regulate the epigenetic silencing of several cancer-related genes, including the HOXD gene [37,38]. Researchers mainly focus on the deregulation of HOTAIR in many forms of cancer due to its remarkable effect on epigenetic regulation at the genome-wide level. In the present study, we analyzed the influence of genetic variations in HOTAIR on the risk of developing CRC in the Saudi population. The increased expression level of HOTAIR is reported in many different types of cancer tissue samples with higher levels in the metastatic stage. Many in vivo and in vitro studies showed the upregulation of HOTAIR expression with enhanced tumor invasion and metastasis [39,40,41]. In CRC patients, many studies revealed a higher expression of HOTAIR in CRC tissue as compared to corresponding noncancerous tissue [42,43,44]. The SNPs rs920778 and rs12826786 are reported to correlate with HOTAIR upregulation [31,32,33]. The AA genotype in rs920778 SNP (G > A) situated in the intronic enhancer region can increase the expression of HOTAIR [45]. Presently, we reported a strong association of rs920778 G > A with increased CRC risk in the Saudi population, and a potential biomarker, the homozygous mutant genotype “AA”, showed a significant association with the risk of CRC, particularly with gender, age, and location. Some previous studies have reported the association of rs920778 SNP with increased cancer risk in Turkish, Indian, and Chinses populations [32,34,45]. Recently, HOTAIR rs920778 polymorphisms with higher survival rates were reported in CRC patients from South Korea [19] and in breast cancer patients in southeast Iran [46].

The AA genotype in HOTAIR rs1899663 SNP C > A situated at the intronic region can alter the affinity for binding of several transcription factors, including paired Box 4, spermatogenic leucine zipper 1, and zinc finger protein 281, resulting in increased expression levels of HOTAIR [47]. We observed a significant protective association of the additive genotype “GA + AA” of HOTAIR rs1899663C > A in CRC susceptibility in Saudi CRC patients, and the minor allele “A” showed a decreased OR in patients who had tumors located in the colon. The homozygous variant genotype “AA” was associated with late-stage tumors (III-IV), and the genotype “CA + AA” showed a significant protective association in younger patients while it had a nonsignificant risk association in patients older than 57 years of age, which suggests that rs1899663 had a protective role in younger Saudi patients. Earlier in the South Korean population, the TT genotype was reported to increase mortality in CRC patients who displayed tumors in the colon region only [19]. Wang et al. [33] reported that rs1899663 (G > T) increased lung cancer risk in China, and Hassanzarei et al. [46] reported a negative association of rs1899663 (G > T) with breast cancer in Iran. Additionally, rs1899663 polymorphism is linked with breast cancer risk in Indian and Chinese populations and prostate cancer susceptibility in the Iranian population [32,34,47].

The genotype TT in rs12826786 can be considered a protective mutation against susceptibility to CRC in the Saudi population as it was associated with a reduced risk of CRC in male patients and patients with advanced tumors. Our results are consistent with Iranian populations [39] where an rs12826786 (C > T) polymorphism showed a protective association with breast cancer. Likewise, Kashani et al. [35] did not find any association between HOTAIR rs12826786 and Lymphoma risk while Hassanzarei et al. [46] reported a decreased risk of breast cancer with rs12826786 polymorphisms.

An LD analysis was performed for these three SNPs (rs920778 G > A, rs12816786 C > T, and rs1899663 C> A) to determine the potential for the nonrandom association of alleles in the Saudi population. The LD pattern among the SNPs was measured using the correlation coefficient D’ (r2). Our results showed LD blocks among the three SNPs selected; the r2 values indicate that these loci are coinherited and that there was a tendency for alleles to be transmitted together. However, there was no clear association between these alleles in the Saudi population. After performing the RNAsnp prediction analysis, the results indicated that the RNA secondary structures of the HOTAIR genotypes were slightly changed, indicating that these SNPs may participate in colorectal cancer via the alteration of the HOTAIR lncRNA secondary structure. This may contribute to the reduced efficiency of their function.

## 5. Conclusions

This study is the first to analyze the association between HOTAIR gene polymorphisms and the prevalence of CRC in the Saudi population. Our results suggested a strong association of rs920778 with increased CRC risk while rs12826786 and rs1899663 showed a protective role in this population. HOTAIR polymorphism can represent a useful biomarker for the early diagnosis of CRC. The size of our sample was small; thus, our findings must be proven in a larger number of samples to improve the scientific rigor. In the present study, we collected information about the patient’s age, gender, tumor stage, and location, but not considering familial CRC history is the main limitation of this study. We hope that this work will help improve the understanding of cancer mechanisms and stimulate the discovery of therapeutic targets that are less harmful than chemotherapy.

## Figures and Tables

**Figure 1 genes-14-00592-f001:**
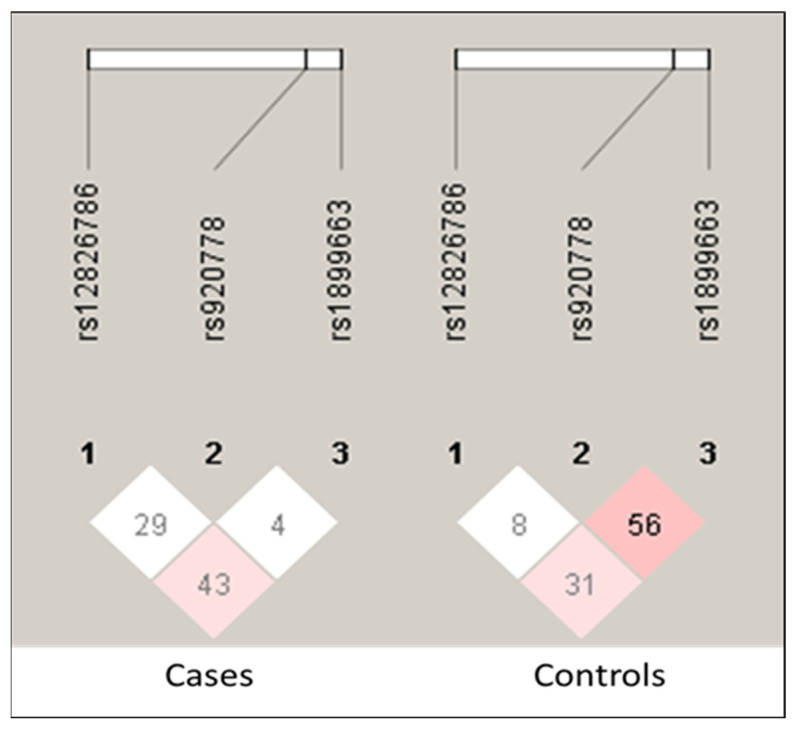
Linkage disequilibrium association among the three HOTAIR SNPs in colorectal cancer and controls. The darker red color indicates a higher r2 value.

**Figure 2 genes-14-00592-f002:**
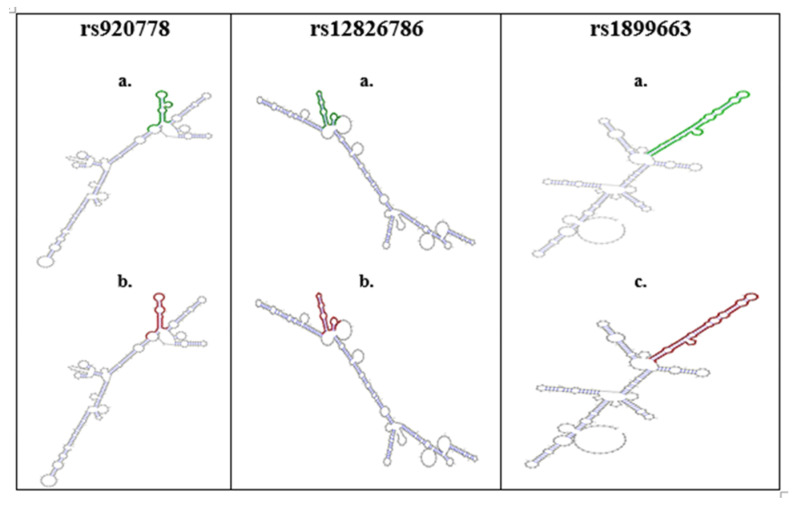
Secondary structure of HOTAIR SNPs and base-pair probabilities. a—wild type; b—mutant type.

**Table 1 genes-14-00592-t001:** Demographic characteristics of the study subjects for genotyping study.

Characteristics	Cases, *n* (%)	Control, *n* (%)
	(*n* = 144)	(*n* = 144)
Age in years (mean ± SD)		
≤57	68 (47%)	78 (54.17%)
>57	76 (52.7%)	66 (45.83%)
Sex		
Males	86 (59.7%)	82 (56.94%)
Females	58 (40%)	62 (43.06%)
Tumor location		
Colon	91 (63%)	
Rectum	53 (36.8%)	
Tumor node metastasis		
Stage I-II	61 (54%)	
Stage III-IV	51 (46%)	

**Table 2 genes-14-00592-t002:** Comparison of polymorphisms in HOTAIR genotype frequencies between CRC cases and healthy controls.

Genotype	Controls	Cases	OR (95% CI)	χ2	*p*
	(*n* = 144)	(*n* = 144)			
rs920778 G > A					
GG	48 (33%)	35 (24%)	1.000 (reference)		
GA	70 (49%)	70 (49%)	1.371 (0.793–2.371)	1.28	0.25741
AA	26 (18%)	39 (27%)	2.057 (1.063–3.981)	4.64	0.03131
GA + AA	96 (67%)	109 (76%)	1.557 (0.931–2.606)	2.86	0.09078
G	83 (58%)	70 (49%)			
A	61 (42%)	74 (51%)	1.438 (1.035–1.998)	4.71	0.02994
rs1899663 C > A					
CC	10 (7%)	16 (11%)	1.000 (reference)		
CA	64 (44%)	7 (5%)	0.068 (0.023–0.208)	28.1	1.153
AA	69 (48%)	58 (40%)	0.525 (0.221–1.246)	2.18	0.14015
CA + AA	133 (92%)	65 (45%)	0.305 (0.131–0.710	8.21	0.00417
C	42 (29%)	19.5 (14%)			
A	101 (70%)	61.5 (43%)	1.312 (0.844–2.038)	1.46	0.22746
rs12826786C > T					
CC	93 (65%)	92 (64%)	1.000 (reference)		
CT	40 (28%)	49 (34%)	1.238 (0.746–2.057)	0.68	0.40872
TT	11 (8%)	3 (2%)	0.276 (0.074–1.020)	4.18	0.04094
CT + TT	51 (35%)	52 (36%)	1.031 (0.637–1.669)	0.02	0.90215
C	133 (92%)	116.5 (81%)			
T	31 (22%)	27.5 (19%)	0.86 (0.573–1.292)	0.53	0.46848

**Table 3 genes-14-00592-t003:** The relation between HOTAIR gene polymorphisms and colorectal cancer risk with age.

Genotype	≤57					>57				
	Controls	Cases	OR (95% CI)	χ2	*p*	Controls	Cases	OR (95% CI)	χ2	*p*
	(*n* = 78)	(*n* = 68)				(*n* = 66)	(*n* = 76)			
rs920778 G > A										
GG	26 (33%)	15 (22%)	1.000 (reference)			22 (33%)	20 (26%)	1.000 (reference)		
GA	31 (40%)	29 (43%)	1.622 (0.720–3.654)	1.4	0.24	39 (59%)	41 (54%)	1.156 (0.548–2.442)	0.15	0.703
AA	21 (27%)	24 (35%)	1.981 (0.835–4.701)	2.4	0.12	5 (8%)	15 (20%)	3.3 (1.015–10.733)	4.13	0.043
GA + AA	52 (67%)	53 (78%)	1.767 (0.841–3.709)	2.3	0.13	44 (67%)	56 (74%)	1.4 (0.679–2.885)	0.84	0.361
G	28.5 (37%)	29.5 (43%)				30.5 (46%)	40.5 (53%)			
A	36.5 (47%)	38.5 (57%)	1.484 (0.934–2.356)	2.8	0.09	24.5 (37%)	35.5 (47%)	1.485 (0.923–2.389)	2.66	0.103
rs1899663 C > A										
CC	4 (5%)	11 (16%)	1.000 (reference)			6 (9%)	5 (7%)	1.000 (reference)		
CA	41 (53%)	26 (38%)	0.231 (0.066–0.801)	5.9	0.0151	23 (35%)	44 (58%)	2.296 (0.632–8.336)	1.65	0.198
AA	33 (42%)	31 (46%)	0.342 (0.098–1.186)	3	0.08	36 (55%)	27 (36%)	0.9 (0.248–3.261)	0.03	0.873
CA + AA	74 (95%)	57 (84%)	0.28 (0.085–0.926)	4.81	0.0289	59 (89%)	71 (93%)	1.444 (0.420–4.970)	0.34	0.558
C	24.5 (31%)	24 (35%)				17.5 (27%)	27 (36%)			
A	53.5 (69%)	44 (65%)	0.84 (0.515–1.367)	0.5	0.48	47.5 (72%)	49 (64%)	0.669 (0.401–1.114)	2.4	0.121
rs12826786C > T										
CC	54 (69%)	45 (66%)	1.000 (reference)			39 (59%)	47 (62%)	1.000 (reference)		
CT	18 (23%)	21 (31%)	1.4 (0.666–2.945)	0.8	0.37	22 (33%)	28 (37%)	1.056 (0.524–2.130)	0.02	0.879
TT	6 (8%)	2 (3%)	0.4 (0.077–2.080)	1.3	0.26	5 (8%)	1 (1%)	0.166 (0.019–1.481)	3.24	0.072
CT + TT	24 (31%)	23 (34%)	1.15 (0.574–2.305)	0.2	0.69	27 (41%)	29 (38%)	0.891 (0.454–1.750)	0.11	0.738
C	63 (81%)	55.5 (82%)				50 (76%)	61 (80%)			
T	15 (19%)	12.5 (18%)	0.946 (0.525–1.705)	0	0.85	16 (24%)	15 (20%)	0.768 (0.437–1.351)	0.84	0.359

**Table 4 genes-14-00592-t004:** The relation between HOTAIR gene polymorphisms and colorectal cancer risk with gender.

Genotype	Males					Females				
	Controls	Cases	OR (95% CI)	χ2	*p*	Controls	Cases	OR (95% CI)	χ2	*p*
	(*n* = 82)	(*n* = 86)				(*n* = 62)	(*n* = 58)			
rs920778 G > A										
GG	29 (35%)	24 (28%)	1.000 (reference)			19 (31%)	11 (19%)	1.000 (reference)		
GA	37 (45%)	42 (49%)	1.37 (0.682–2.758)	0.79	0.375	33 (53%)	28 (48%)	1.466 (0.598–3.595)	0.7	0.403
AA	16 (20%)	20 (23%)	1.51 (0.645–3.538)	0.91	0.341	10 (16%)	19 (33%)	3.282 (1.129–9.536)	4.91	0.0266
GA + AA	53 (65%)	62 (72%)	1.41 (0.735–2.717)	1.08	0.298	43 (69%)	47 (81%)	1.888 (0.807–4.417)	2.18	0.14
G	47.5 (58%)	45 (52%)	Ref			35.5 (57%)	15 (26%)			
A	34.5 (42%)	41 (48%)	1.25 (0.815–1.930)	1.06	0.302	26.5 (43%)	33 (57%)	1.768 (1.060–2.949)	4.8	0.0284
rs1899663 C > A										
CC	6 (7%)	8 (9%)	1.000 (reference)			4 (6%)	8 (14%)	1.000 (reference)		
CA	33 (40%)	37 (43%)	0.84 (0.264–2.677)	0.09	0.769	31 (50%)	33 (57%)	0.532 (0.146–1.946)	0.93	0.335
AA	42 (51%)	41 (48%)	0.73 (0.234–2.295)	0.29	0.592	27 (44%)	17 (29%)	0.315 (0.082–1.208)	3	0.083
CA + AA	75 (91%)	78 (91%)	0.78 (0.258–2.355)	0.2	0.659	58 (94%)	50 (86%)	0.431 (0.122–1.517)	1.79	0.18
C	22.5 (27%)	26.5 (31%)				19.5 (31%)	24.5 (42%)			
**A**	58.5 (71%)	59.5 (69%)	0.86 (0.539–1.385)	0.37	0.542	13.5 (22%)	33.5 (58%)	0.627 (0.370–1.064)	3	0.083
**rs12826786C > T**										
**CC**	56 (68%)	61 (71%)	1.000 (reference)			37 (60%)	31 (53%)	1.000 (reference)		
CT	18 (22%)	23 (27%)	1.17 (0.574–2.399)	0.19	0.662	22 (35%)	26 (45%)	1.411 (0.672–2.961)	0.83	0.363
TT	8 (10%)	2 (2%)	0.23 (0.047–1.127)	3.81	0.05107	3 (5%)	1 (2%)	0.398 (0.039–4.020)	0.65	0.421
CT + TT	26 (32%)	25 (29%)	0.88 (0.457–1.704)	0.14	0.71	25 (40%)	27 (47%)	1.289 (0.625–2.658	0.47	0.491
C	65 (79%)	72.5 (84%)				48 (77%)	44 (76%)			
T	17 (21%)	13.5 (16%)	0.71 (0.408–1.244)	1.43	0.231	14 (23%)	13 (22%)	1.091 (0.600–1.985)	0.08	0.776

**Table 5 genes-14-00592-t005:** The relation between HOTAIR gene polymorphisms and colorectal cancer risk with tumor location.

Genotype	Control	Colon	OR (95% CI)	χ2	*p*	Rectum	OR (95% CI)	χ2	*p*
	(*n* = 144)	(*n* = 91)				(*n* = 53)			
rs920778 G > A									
GG	48 (33%)	19 (21%)	1.000 (reference)			16 (30%)	1.000 (reference)		
GA	70 (49%)	48 (53%)	1.7 (0.908–3.305)	2.81	0.0938	22 (42%)	0.9 (0.449–1.979)	0.02	0.403
AA	26 (18%)	24 (26%)	2.3 (1.082–5.027)	4.75	0.02926	15 (28%)	1.7 (0.739–4.053)	1.61	0.0266
GA + AA	96 (67%)	72 (79%)	1.8 (1.027–3.497)	4.24	0.04209	37 (70%)	1.15 (0.585–2.285)	0.17	0.14
G	83 (58%)	43 (47%)				27 (51%)			
A	61 (42%)	48 (53%)	1.51 (1.046–2.206)	4.84	0.02785	26 (49%)	1.3 (0.838–2.048)	1.41	0.0284
rs1899663 C > A									
CC	10 (7%)	16 (11%)	1.000 (reference)			4 (8%)	1.000 (reference)		
CA	64 (44%)	7 (5%)	0.58 (0.233–1.473)	1.31	0.2525	25 (47%)	0.97 (0.280–3.403)	0	0.97
AA	69 (48%)	58 (40%)	0.41 (0.161–1.045)	3.61	0.05726	25 (47%)	0.87 (0.260–3.151)	0.02	0.88
CA + AA	133 (92%)	65 (45%)	0.49 (0.204–1.198)	2.5	0.1135	50 (94%)	0.91 (0.282–3.134)	0.01	0.92
C	42 (29%)	19.5 (14%)				16.5 (31%)			
A	101 (70%)	61.5 (43%)	0.68 (0.460–1.009)	3.69	0.05482	37.5 (71%)	0.9 (0.584–1.530)	0.05	0.82
rs12826786C > T									
CC	93 (65%)	92 (64%)	1.000 (reference)			35 (66%)	1.000 (reference)		
CT	40 (28%)	49 (34%)	1.3 (0.738–2.308)	0.84	0.3591	17 (32%)	1.129 (0.568–2.247)	0.12	0.73
TT	11 (8%)	3 (2%)	0.29 (0.063–1.387)	2.65	0.1036	1 (2%)	0.242 (0.030–1.941)	2.08	0.15
CT + TT	51 (35%)	52 (36%)	1.08 (0.631–1.876)	0.09	0.7623	18 (34%)	0.93 (0.483–1.820)	0.04	0.85
C	133 (92%)	116.5 (81%)				43.5 (82%)			
T	31 (22%)	27.5 (19%)	0.89 (0.567–1.424)	0.21	0.6496	9.5 (18%)	0.79 (0.450–1.408)	0.62	0.43

**Table 6 genes-14-00592-t006:** The relation between HOTAIR gene polymorphisms and colorectal cancer risk with tumor stage.

Genotype	Controls	Stage I-II (*n* = 61)	OR (95% CI)	χ2	*p*	Stage I-II	OR (95% CI)	χ2	*p*
	(*n* = 144)					(*n* = 51)			
rs920778 G > A									
GG	48 (33%)	12 (20%)	1.000 (reference)			13 (25%)	1.000 (reference)		
GA	70 (49%)	32 (52%)	1.829 (0.857–3.90)	2.47	0.11606	22 (43%)	1.16 (0.533–2.526)	0.14	0.70757
AA	26 (18%)	17 (28%)	2.615 (1.085–6.304)	4.73	0.02972	16 (31%)	2.272 (0.948–5.44)	3.46	0.06272
GA + AA	96 (67%)	49 (80%)	2.042 (0.994–4.195)	3.86	0.04937	38 (75%)	1.462 (0.712–2.999)	1.08	0.2992
G	83 (58%)	28 (46%)				24 (47%)			
A	61 (42%)	33 (54%)	1.604 (1.047–2.455)	4.76	0.02921	27 (53%)	1.531 (0.972–2.499)	3.4	0.065
rs1899663 C > A									
CC	10 (7%)	4 (7%)	1.000 (reference)			8 (16%)	1.000 (reference)		
CA	64 (44%)	26 (43%)	1.016 (0.292–3.530)	0	0.98054	24 (47%)	0.469 (0.165–1.328)	2.09	0.14819
AA	69 (48%)	31 (51%)	1.123 (0.327–3.860)	0.03	0.85361	19 (37%)	0.344 (0.119–0.993)	4.11	0.0426
CA + AA	133 (92%)	57 (93%)	1.071 (0.323–3.558)	0.01	0.91029	43 (84%)	0.0404 (0.150–1.089)	3.38	0.06619
C	42 (29%)	17 (28%)				20 (39%)			
A	101 (70%)	44 (72%)	1.076 (0.672–1.723)	0.09	0.75937	31 (61%)	0.645 (0.402–1.033)	3.35	0.06715
rs12826786C > T									
CC	93 (65%)	43 (70%)	1.000 (reference)			32 (63%)	1.000 (reference)		
CT	40 (28%)	17 (28%)	0.919 (0.469–1.801)	0.06	0.80605	19 (37%)	1.38 (0.701–2.719)	0.87	0.3503
TT	11 (8%)	1 (2%)	0.197 (0.025–1.572)	2.86	0.09071	0%	0.125 (0.007–2.183)	3.68	0.05499
CT + TT	51 (35%)	18 (30%)	0.763 (0.399–1.459)	0.67	0.41309	19 (37%)	1.083 (0.558–2.100)	0.06	0.81408
C	133 (92%)	51.5 (84%)				41.5 (81%)			
T	31 (22%)	9.5 (16%)	0.672 (0.382–1.182)	1.92	0.16626	9.5 (19%)	0.834 (0.471–1.479)	0.39	0.53492

## Data Availability

The data are presented in the current manuscript.

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
