# Peer review of "Genetic Variants of HOTAIR Associated with Colorectal Cancer: A Case-Control Study in the Saudi Population"

_genes, 2023, doi:10.3390/genes14030592_

Round 1

Reviewer 1 Report

By screening the SNP results of 144 CRC cases and 144 controls, the authors find that a significant association between SNP rs920778 G>A and CRC risk and a protective role of SNPs rs12826786 (C>T) and rs1899663 (C>A) was noticed.

However, I have some concerns about the results.

1. The frequency of rs920778 G>A is 61 (42%) in the controls V.S. 74 (51%) in the cases (OR (95% CI): 1.438(1.035–1.998). Since the difference is not significant in the frequencies of 42% in controls and 51% in cases. Maybe it is better to enroll more samples to get a more robust result.

2. It is better to study the relation between HOTAIR gene polymorphisms and colorectal cancer risk according to familial cancer history, especially familial CRC history. 

Author Response

Reviewer 1

Comments and Suggestions for Authors

By screening the SNP results of 144 CRC cases and 144 controls, the authors find that a significant association between SNP rs920778 G>A and CRC risk and a protective role of SNPs rs12826786 (C>T) and rs1899663 (C>A) was noticed.

However, I have some concerns about the results.

1.The frequency of rs920778 G>A is 61 (42%) in the controls V.S. 74 (51%) in the cases (OR (95% CI): 1.438(1.035–1.998). Since the difference is not significant in the frequencies of 42% in controls and 51% in cases. Maybe it is better to enroll more samples to get a more robust result.

Thank you for your valuable comment. Unfortunately, it will be very difficult to enroll more participants in the study right now. We will definitely recruit a large number of participants in our future studies

  1. It is better to study the relation between HOTAIR gene polymorphisms and colorectal cancer risk according to familial cancer history, especially familial CRC history.

Familial CRC history was not included in the study design.  But we will definitely include it in all our future studies.

Reviewer 2 Report

1. 4 ml blood in which tube missing properly mention 4 ml blood in EDTA or any other blood container tube and what is the company of the tube for example BD vacutainer with country USA or what?

2. SNPs were selected on which basis?

3. No. of cycles of the PCR missing, for example, repeated cycles 36 or what?

4. IBM SPSS was used but missing its citation and version.

5. Discussion and conclusion need improvement.

Author Response

Reviewer 2

Comments and Suggestions for Authors

  1. 4 ml blood in which tube missing properly mention 4 ml blood in EDTA or any other blood container tube and what is the company of the tube for example BD vacutainer with country USA or what?

Done

  1. SNPs were selected on which basis?

These SNPs were selected on the basis of some recent reports linking them with increased cancer risk. References were added in the introduction

  1. No. of cycles of the PCR missing, for example, repeated cycles 36 or what?

Revised

  1. IBM SPSS was used but missing its citation and version.

Updated

  1. Discussion and conclusion need improvement.

Done

Reviewer 3 Report

A brief summary

This study aimed to determine the association of three single nucleotide polymorphisms in the HOTAIR as long non coding RNA gene in colorectal cancer in the Saudi Arabia. The authors have shown that polymorphism rs920778 is associated with the risk of colorectal cancer, but two polymorphisms, rs12826786 and rs1899663, have a protective role for colon cancer.

General concept comment :

The major points that should be adresse are :

The basis of the selection of polymorphisms is not given in the text, but investigating the function of polymorphisms on the secondary structure of non-coding RNA is one of the strengths of this study, although it would be better to do this before selecting polymorphisms.

Specific comments (might be redundant with the general comments for some points (

1. In the abstract section, it is better to mention the method by which you determined the genotype of the polymorphisms

2. The authors have first investigated the relationship between polymorphisms and colorectal cancer, then they have discussed whether these polymorphisms have an effect on the secondary structure of HOTAIR or not. Wouldn't it be better to choose these polymorphisms based on the effect on the secondary structure?

Author Response

Reviewer 3

 Genetic Variants of HOTAIR associated with Colorectal Cancer: A Case-Control Study in the Saudi Population

A brief summary

This study aimed to determine the association of three single nucleotide polymorphisms in the HOTAIR as long non coding RNA gene in colorectal cancer in the Saudi Arabia. The authors have shown that polymorphism rs920778 is associated with the risk of colorectal cancer, but two polymorphisms, rs12826786 and rs1899663, have a protective role for colon cancer.

General concept comment:

The major points that should be addressed are:

The basis of the selection of polymorphisms is not given in the text, but investigating the function of polymorphisms on the secondary structure of non-coding RNA is one of the strengths of this study, although it would be better to do this before selecting polymorphisms.

These SNPs were selected on the basis of some recent reports linking them with increased cancer risk. We have updated the  references in the introduction part

Specific comments (might be redundant with the general comments for some points

  1. In the abstract section, it is better to mention the method by which you determined the genotype of the polymorphisms

Revised

  1. The authors have first investigated the relationship between polymorphisms and colorectal cancer, then they have discussed whether these polymorphisms have an effect on the secondary structure of HOTAIR or not. Wouldn't it be better to choose these polymorphisms based on the effect on the secondary structure?

I totally agree with the suggestion and will imply it in our future studies
